# Characterization of a Phage-Encoded Depolymerase Against *Klebsiella pneumoniae* K30 Capsular Type and Its Therapeutic Application in a Murine Model of Aspiration Pneumonia

**DOI:** 10.3390/v17111446

**Published:** 2025-10-30

**Authors:** Yuchen Chen, Zheng Fan, Tongtong Fu, Zhoufei Li, Junxia Feng, Xiaohu Cui, Lin Gan, Guanhua Xue, Yanling Feng, Hanqing Zhao, Jinghua Cui, Chao Yan, Ziying Xu, Zihui Yu, Yang Yang, Yuehua Ke, Jing Yuan

**Affiliations:** 1Department of Bacteriology, Capital Institute of Pediatrics, Beijing 100020, China; chenyuchen1003cyc@gmail.com (Y.C.); fanzheng_123@163.com (Z.F.); dtongfu@126.com (T.F.); lizhf35@163.com (Z.L.); fengjx614@126.com (J.F.); 15989001667@163.com (X.C.); ganlin0622@126.com (L.G.); xgh618@163.com (G.X.); fengyanling9991@sina.com (Y.F.); lights012@163.com (H.Z.); cuijinghua7910@hotmail.com (J.C.); cip_yanchao@163.com (C.Y.); xuziying326@126.com (Z.X.); yuzihui1230@163.com (Z.Y.); yyang920111@163.com (Y.Y.); yuehuakebj@163.com (Y.K.); 2Capital Institute of Pediatrics, Chinese Academy of Medical Sciences & Peking Union Medical College, Beijing 100730, China

**Keywords:** *Klebsiella pneumoniae*, capsular polysaccharide, phage, depolymerase

## Abstract

Capsular polysaccharides are critical virulence factors of *Klebsiella pneumoniae*, enabling the bacterium to evade host immune recognition and exacerbate infection. Phage-derived depolymerases, which specifically degrade these capsular polysaccharides, are increasingly recognized as a highly promising strategy for the treatment of bacterial infections. In the present study, we isolated and characterized a lytic *Klebsiella pneumoniae* phage, named phiTH1, and sequenced its genome. The K30-type capsular polysaccharide was identified as the receptor for phiTH1 infection. A tail fiber protein with a pectate lyase domain, Dop5, was then recognized as a potential K30-type depolymerase. Therefore, the recombinant protein Dop5 was expressed in *Escherichia coli* and purified, and its in vitro capsular depolymerase activity was demonstrated. Further, by using a murine aspiration pneumonia model induced by K30-type *Klebsiella pneumoniae* TH1, we found that Dop5 protected 80% of mice from lethal challenge with *Klebsiella pneumoniae.* After Dop5 treatment, the pathological damage in multiple organs of mice was alleviated, the bacterial load was reduced, and serum levels of inflammatory cytokines and complement C3 decreased, along with a significant reduction in the pathological score of the lungs. Hence, this study revealed the potential of the depolymerase Dop5 for the treatment of *Klebsiella pneumoniae* infections.

## 1. Introduction

*Klebsiella pneumoniae* is a common Gram-negative pathogen widely distributed in the environment that causes various infections, including pneumonia, urinary tract infections, bacteremia, and pyogenic liver abscesses [1,2]. The primary virulence determinant of *K. pneumoniae* is a thick capsular polysaccharide (CPS) that coats the bacterial surface and helps the pathogen evade host immune clearance [3]. CPS can protect bacteria from complement-mediated bactericidal activity and phagocytosis, inhibit early inflammatory responses, and promote the development of severe infections. Moreover, the polysaccharide matrix of CPS can hinder antibiotic penetration, leading to treatment failure [4]. The recent emergence and spread of multidrug-resistant and hypervirulent *K. pneumoniae* phenotypes poses a significant threat to public health, underscoring the urgent need for novel antibacterial strategies to control such infections.

Bacteriophages can specifically recognize target bacteria and achieve precise sterilization without disturbing the normal host flora, making them a potential alternative treatment for antibiotic-resistant infections [5,6]. However, high host specificity and a narrow antibacterial spectrum limit their clinical application, particularly in the treatment of rare or mixed bacterial infections [7]. Notably, bacteria can rapidly produce phage-resistant variants through receptor mutations under the selective pressure of phage therapy. These mutations hinder phage adsorption and infection, greatly reducing their therapeutic effects [8]. Most studies have solved this problem by screening new phages or by using a combination of multiple phages; however, this increases the treatment complexity [9,10].

As functional components of bacteriophages, depolymerases participate in the cleavage of bacterial surface CPS [11], promoting the further binding of phages to bacterial receptors to complete the infection process [12]. In addition to the direct application of phages, depolymerase also shows potential for independent treatment as an antibacterial drug [13]. Depolymerases have a more stable and consistent effect on strains of the same capsule type than bacteriophages [14]. Moreover, capsule degradation by depolymerases can reduce bacterial virulence, enhance immune recognition, and promote phagocytosis and complement-mediated killing by host immune cells [15]. To date, a variety of depolymerases have been successfully identified, targeting at least 25 different capsule serotypes, such as K1, K2, K5, K8, K20, K27, K47, and K56, providing a new approach for the personalized treatment of different *K. pneumoniae* infections [16].

In this study, a novel phage, phiTH1, which can lyse K30-type *K. pneumoniae* TH1, was isolated and identified. Based on phenotypic and genomic analyses, we identified the depolymerase Dop5, encoded by the *orf5* gene of phiTH1, which specifically degrades the CPS of *K. pneumoniae* TH1. To evaluate the therapeutic effect of Dop5, we established a murine aspiration pneumonia model infected with *K. pneumoniae* TH1. Our results demonstrate that Dop5 provides significant protection in infected mice, highlighting its potential as a promising antimicrobial agent.

## 2. Materials and Methods

### 2.1. Bacterial Strains and Culture Conditions

All *K. pneumoniae* strains used in this study were sourced from our laboratory and cultivated in fresh Luria–Bertani (LB) broth (Beijing Land Bridge Technology Co., Ltd., Beijing, China, catalog number: CM158). As previously described, a K30-type *K. pneumoniae* TH1 isolate was isolated from a patient with severe nonalcoholic steatohepatitis (NASH) accompanied by auto-brewery syndrome (ABS) [17]. *K. pneumoniae* TH1 was designated as the host for phage isolation and propagation. Δ*wza* was constructed based on previous experimental methods by Link et al. [18]. Details of all strains, plasmids and primers used in this study are presented in Appendix A.

### 2.2. Isolation and Purification of Phage PhiTH1

Phage phiTH1 was isolated from untreated sewage at the Capital Institute of Pediatrics (Beijing, China). The isolation and purification of bacteriophages were performed by the double-layer agar method as previously described [19]. In brief, 20 mL of untreated sewage was subjected to centrifugation at 4400 rpm for 20 min. The resulting supernatant was filtered and sterilized with a 0.22 µm microfiltration membrane (Pall Corporation, New York, NY, USA) to obtain the initial phage suspension. After 100 µL of the phage suspension was mixed with 100 µL of *K. pneumoniae* TH1 and incubated overnight, the supernatant was collected by centrifugation. The 100 µL of logarithmic-phase *K. pneumoniae* TH1 was then mixed, respectively, with 100 µL of ten-fold serially diluted phage suspension in 5 mL sterile tubes. After incubation for 15 min, 4 mL of LB semi-solid medium was added to each mixture, which was then thoroughly mixed and poured onto the surface of LB solid medium. The plates were inverted and incubated for 16–18 h at 37 °C. Subsequently, individual phage plaques were incubated in 1 mL SM buffer overnight, then eluted with shaking at 37 °C for 4 h. This procedure was repeated several times until homogenous looking plaques were obtained to ensure phage purity.

### 2.3. Plaque Formation and Quantification of Phage PhiTH1

The dimensions of the plaques were quantified using ImageJ software (Version 1.54p) [20]. Each plaque was considered equivalent to a single plaque-forming unit (PFU), and phage titers were expressed in PFU/mL. To ensure precision, the measurements were performed in triplicate.

### 2.4. Transmission Electron Microscopy (TEM)

Cesium chloride (CsCl) density gradient ultracentrifugation was performed to purify the phage particles. Four CsCl solutions with densities of 1.33, 1.45, 1.50 and 1.70 g/mL were prepared in SM buffer. The SW55Ti swinging-bucket rotor (Beckman Coulter, Brea, CA, USA) was pre-cooled to 4 °C. A discontinuous CsCl gradient was formed in a 5 mL ultracentrifuge tube through the sequential addition and layering of 1.1 mL of each solution using a syringe. Subsequently, 0.5 mL of the crude phage lysate was gently loaded onto the top of the gradient. The tubes were balanced and centrifuged at 140,000× *g* for 3 h at 4 °C. Following centrifugation, the tube was fixed upright, and the viral band was visualized under strong light. The band was collected by piercing the tube wall just below it with a 1 mL sterile syringe and carefully aspirated. The phage suspension was transferred to a Slide-A-Lyzer dialysis cassette (10,000 MWCO; Thermo Fisher Scientific, Rockford, IL, USA). The cassette was then subjected to dialysis against 3 L of cold phosphate-buffered saline (PBS) (pH 7.6) in a 4 L beaker at 4 °C with continuous gentle stirring using a magnetic stirrer. The dialysis buffer was replaced every 2–3 h, and this process was continued for a total of 12 h. The infectivity of phage suspension was verified by a spot assay with the host bacteria. Phage morphology was examined by TEM using standard negative-staining techniques with 2% uranyl acetate.

The bacterial capsule was observed by transmission electron microscopy. A 20 mmol/L HEPES buffer solution was prepared by diluting 1 mol/L HEPES buffer solution. Overnight bacterial cultures were transferred and adjusted to an optical density at 600 nm (OD_600_) of 0.5 using a spectrophotometer (Shanghai Metash Instruments Co., Ltd., Shanghai, China). Bacterial pellets were harvested by centrifugation at 12,000 rpm for 5 min and then resuspended in 1 mL HEPES buffer. Subsequently, 80 µL of ferritin was added to the suspension, and the mixture was incubated at room temperature for 30 min. The sample was then subjected to centrifugation at 12,000 rpm for 15 min. The resultant pellets were washed twice with 1 mL HEPES buffer to remove any excess ferritin. After the supernatant was discarded, 2.5% glutaraldehyde fixing solution was gently added along the tube wall for electron microscopy.

### 2.5. The Optimal Multiplicity of Infection (MOI) of Phage PhiTH1

The Optimal multiplicity of infection (MOI) of phage phiTH1 was measured as previously described [21]. At MOIs of 0.00001, 0.0001, 0.001, 0.01, 0.1, 1, 10, and 100, ten-fold serially diluted phage suspension to be tested were mixed with logarithmic-phase *K. pneumoniae* TH1. The mixtures were incubated overnight at 37 °C with constant shaking at 200 rpm, and then subjected to centrifugation at 12,000 rpm for 5 min. The resulting supernatants were then collected and filtered using 0.22 µm microfiltration membrane (Pall Corporation, New York, USA) to remove bacteria. The double-layer agar method was performed to measure the phage titers, with the highest titer of infection being considered the optimal MOI. Each experiment was repeated three times.

### 2.6. One-Step Growth Curve of Phage PhiTH1

The one-step growth curve was established to analyze phage replication kinetics according to a reported method [22]. Briefly, phage phiTH1 was added to logarithmic-phase *K. pneumoniae* TH1 at the optimal MOI and incubated for 10 min at 37 °C. Unabsorbed phages were removed by centrifugation at 12,000× *g* for 5 min and the resulting pellets were resuspended in 20 mL of LB medium. The phage suspension was then incubated for 150 min at 37 °C with shaking at 200 rpm. The samples were obtained at 10 min intervals and the titers of phages were assayed by the double-layer agar method. To calculate the phage burst size, the phage titers at the post-burst plateau phase were divided by the initial titers. The experiment was conducted with three replications.

### 2.7. Evaluation of Phage Stability Under Different Thermal and pH Conditions

For the evaluation of phage stability at different pH levels and temperatures, phage suspensions were incubated at different temperatures (4, 10, 20, 30, 40, 50, 60, 70, 80, 90, and 100 °C) and at a pH range from 2 to 13 for 1 h. Phage titers were determined by the double-layer agar method.

### 2.8. Phage Sensitivity Testing

The double-layer agar method was used to measure the titers of phage phiTH1. We then prepared serial dilutions of phage suspensions covering a range from 10^1^ to 10^6^ PFU/mL. Subsequently, 100 µL of overnight bacterial cultures were taken and mixed with 4 mL of LB semi-solid medium. The mixtures were poured onto the surface of a solid LB medium. After the semi-solid layer solidified within 15 min, a 1 µL aliquot from each dilution of the serial dilution series was spotted onto the surface of the double-layer agar plates. The plates were then incubated at 37 °C overnight, after which the plaque formation was observed and recorded. In addition, phage lytic activity can be preliminarily assessed by observing whether the bacterial cultures become clear following co-incubation with the phage.

### 2.9. Genome Sequencing and Analysis

DNA was extracted from phages using the phenol-chloroform method as previously described [23]. Whole-genome sequencing was performed using the Illumina NovaSeq 6000 platform. The resulting genome was compared to public databases using BLASTn 2.17.0+. Potential open reading frames (ORFs) were identified using PHASTER [24], and putative proteins were compared to proteins in the NCBI GenBank database via BLASTp. Circular genomic maps of the phage genomes were conducted using Proksee. Virulence Factor Database (VFDB) was use to predict virulence genes [25]. The prediction of antibiotic resistance genes was carried out by the Resfinder 4.0 [26]. The phylogenomic relationship of phage phiTH1 was analyzed using VipTree 4.0 based on whole-genome comparisons. Phylogenetic analyses were performed using MEGA 12.0.11, applying the Maximum Likelihood method with 1000 bootstrap replicates, based on gene sequences of the predicted depolymerase. Multiple sequence alignments for phylogenetic analysis and secondary structure visualization were generated using ESPript 3.0. The three-dimensional structural models were established by Alphafold3 and plotted using PyMOL 2.6.1. The InterPro database was used to search for target domains, and the Pectin-related functional domain was identified based on the results combined with PyMOL analysis.

### 2.10. Expression and Purification of the Putative Depolymerases

The putative depolymerase fragments were amplified using polymerase chain reaction (PCR) and the PCR products, digested with NdeI and HindIII restriction enzymes, were cloned into the pET-28a expression vector. The expression vector was then transformed into *Escherichia coli* BL2 competent cells. Protein expression was induced using 0.1 mM isopropyl β-D-1-thiogalactopyranoside (IPTG), and the culture was transferred to 16 °C overnight. Bacterial precipitate was collected, and the overexpressed protein was released by sonication. The His-tagged protein was purified by using nickel-affinity chromatography. The concentration of the purified proteins was measured using a BCA protein assay kit, and protein purity and size were verified by sodium dodecyl sulfate-polyacrylamide gel electrophoresis (SDS-PAGE).

### 2.11. Verification of Depolymerase Activity

A volume of 100 µL logarithmic-phase *K. pneumoniae* TH1 was mixed with 4 mL of LB semi-solid medium and poured on solid LB medium. Subsequently, 5 µL of depolymerase protein at different concentrations (80, 40, 20, 10, 5, and 2 ng/µL) was dropped carefully onto the surface of the double-layer agar plates. The plates were then incubated overnight at 37 °C. The assessment of the depolymerization activity were determined by observing the spots.

### 2.12. Quantitation of Bacterial Capsular Polysaccharides

Uronic acid content was quantified according to a previous method [27,28,29]. Bacterial cultures were grown in LB broth at 37 °C with shaking at 200 rpm. After incubation, 5 mL of the culture was harvested by centrifugation at 12,000 rpm for 15 min at 4 °C. For depolymerase treatment, 100 µL of purified enzyme was added to the bacterial pellet, followed by incubation at 37 °C for 1 h. The cells were then washed with 1 mL of sterile distilled water and resuspended in 250 µL of water. Subsequently, 50 µL of 1% Zwittergent 100 mM citric acid solution was added to the suspension, and the mixture was incubated at 50 °C for 30 min. After a brief centrifugation (12,000 rpm, 2 min), the supernatant was transferred and mixed with 1 mL of anhydrous ethanol and left on ice for 1 h. The mixture was then centrifuged again at 12,000 rpm for 5 min at 4 °C. The supernatant was discarded, and the pellet was dissolved in 100 µL of 100 mM HCl. Next, 100 µL of the resuspended sample was combined with 600 µL of sodium tetraborate sulfuric acid and heated at 100 °C for 15 min. After cooling to room temperature, 20 µL of 0.125% (*w*/*v*) carbazole was added, followed by another 15 min incubation at 100 °C. The absorbance was read at 520 nm, and uronic acid concentration was determined using a standard curve.

### 2.13. Growth Curve and Time-Dependent Killing Assay

The antibacterial activity of depolymerase was evaluated through bacterial growth curve and time-dependent killing assay as previously described [30]. Overnight incubated bacterial suspensions were diluted 1:100 into 20 mL LB broth in 50 mL tubes and divided into two experimental groups: TH1, TH1 + Dop5 (10 µg/mL). The cultures were incubated at 37 °C with shaking at 200 rpm. At predetermined time points (0, 2, 4, 6, 12, and 24 h), 200 µL of the samples were taken from the bacterial suspensions to measure its absorbance at 600 nm by microplate reader.

Furthermore, at selected time points (0, 2, 4, 6, 12, and 24 h), a volume of 20 µL of bacterial suspensions was taken and serially diluted ten-fold with LB. A 10 µL aliquot from each dilution of the serial dilution series was dropped on solid LB medium and incubated at 37 °C for 16–18 h for bacterial enumeration. The experiments were conducted with three replications.

### 2.14. Screening and Identification of Phage-Resistant K. pneumoniae Mutants

100 µL of logarithmic-phase *K. pneumoniae* was mixed with 100 µL of a capsule-targeted phage suspensions (10^9^ PFU/mL). The mixture was added to 4 mL of LB semi-solid medium and poured on the LB solid medium. After the top layer solidified, the plates were incubated at 37 °C for 12–18 h. Putative resistant mutants were identified by the presence of single colonies growing on the top agar layer without lysis plaques. These putative mutants were purified by streaking on LB agar plates without phage for three consecutive rounds. After each round of purification, a single colony was selected for the next round of purification. After three rounds of purification, the stable resistance of the purified isolates was confirmed using a standard double-layer agar method.

Following confirmation of stable resistance, genomic DNA from the purified mutants was extracted and subjected to whole-genome sequencing on an Illumina platform. The resulting sequencing reads were aligned to the reference genome using BWA (v0.7.17) (Parameter: mem -t 4 -k 32 -M) [31]. Alignment results were deduplicated using SAMTOOLS (Parameter: rmdup). Individual SNPs were detected using SAMTOOLS (Parameters: -C 50 -mpileup -m 2 -F 0.002 -d 1000) [32] with filtering criteria of a minimum read support of 4 and a minimum quality score (MQ) of 20. Small insertions and deletions (InDels) less than 50 bp in length were detected using SAMTOOLS (Parameters: mpileup -m 2 -F 0.002 -d 1000). Copy number variations (CNVs) were identified using CNVnator (Parameters: -call 100) [33]. BreakDancer software 1.4.4 (minimum confidence score = 80) detected structural variations between the sample and the reference genome based on the alignment of pair-end reads to the reference genome and the actual insert size. All structural variations (SVs) supported by fewer than two paired-end reads were filtered out. Functional annotation of detected variants was performed using ANNOVAR 2015Dec14.

### 2.15. A Murine Model of Pneumonia via Aerosol Inhalation

Male C57BL/6J mice, aged 6–8 weeks, were acquired from Vital River Laboratory Animal Technology Co., Ltd. (Beijing, China). All animals were treated humanely in accordance with the National Institutes of Health guidelines for the ethical use of experimental animals. Based on previous studies, the aerosolized inhalation model has improved acute pneumonia research in mice by simulating natural infection routes, ensuring uniform pathogen distribution, enabling quick infection onset, minimizing invasiveness, and facilitating diverse pathogen and treatment studies [34].

A total of 80 mice were used and divided into two experimental batches: the first batch of 40 mice (*n* = 10 per group) was allocated to survival analysis, while the second batch of 40 mice (*n* = 10 per group) was used for pathological examination, bacterial quantification in organs, and serum analysis. The animals were housed in individually ventilated cages with free access to standard food and antibiotic-free water. Mice were divided into four groups: (1) the control group received normal saline; (2) the TH1 group received 2.5 × 10^8^ CFU of *K. pneumoniae* TH1, followed by normal saline 1 h later; (3) the TH1 + Dop5 group received 2.5 × 10^8^ CFU of *K. pneumoniae* TH1, followed by 50 µg depolymerase Dop5 1 h later; (4) the Dop5 group received normal saline, followed by 50 µg depolymerase Dop5 1 h later. Both the infectious dose for establishing the lethal pneumonia model and the therapeutic dose of Dop5 were determined through preliminary dose-ranging studies conducted prior to the main experiments (Appendix A).

For the first batch, survival was monitored for 7 days, with observations recorded every 12 h. For the second batch, at 24 h post-infection, blood was collected from the orbit to obtain serum for biomarker analysis. Enzyme-linked immunosorbent assay (ELISA) kits (4A Biotech, Suzhou, China) were used to measure the concentrations of inflammatory cytokines (IL-1β, IL-6, and TNF-α) in murine serum. Serum complement C3 levels were measured using a complement C3 assay kit (Szybio Biotech, Wuhan, China, Cat. No. SC16A0200) on a Mindray BS-370E automatic biochemistry analyzer (Mindray, Shenzhen, China). All assays were performed strictly in accordance with the instrument operating procedures and kit instructions. The animals were then euthanized to collect organs including the liver, spleen, lungs, and kidneys. Organs from five mice in each group were homogenized in 500 μL of sterile 1× PBS. The resulting homogenate was serially diluted 10-fold. 10 μL of each dilution (including the undiluted sample) was then spotted on LB agar plates and incubated for bacterial load. The lower limit of detection of this assay was 50 CFU/organ. Tissues from three mice were subjected to histopathological analysis using hematoxylin and eosin (H&E) staining. Lung injury was evaluated based on a standardized histopathological scoring system described in previous studies [21].

### 2.16. Statistical Analysis

Data are presented as mean ± standard deviations (SD) and were analyzed by Student’s *t*-test, one-way analysis of variance (ANOVA), and two-way ANOVA using GraphPad Prism 10.3.0 (GraphPad Software, San Diego, CA, USA). Differences were considered statistically significant at the following *p* values: <0.05 (*), <0.01 (**), <0.001 (***), and <0.0001 (****).

## 3. Results

### 3.1. Isolation of a Novel Phage Hosted by K30-Type K. pneumoniae

The novel phage phiTH1, targeting *K. pneumoniae* TH1 (K30-type), formed large and translucent plaques with a diameter of approximately 0.3 cm surrounded by enlarged halos measuring 0.2 cm (Figure 1A). TEM revealed that the phage possesses an icosahedral head approximately 55–60 nm in diameter and a short, non-contractile tail approximately 13 nm in length, with morphology and dimensions similar to those of bacteriophage T7 (Figure 1B). As shown in Figure 1C, the phage titers reached the highest at an MOI of 0.001. The one-step growth curve indicated that phage phiTH1 had a latent period of 20 min and a burst size of 217 phages per infected cell (Figure 1D). Phage phiTH1 was stable at temperatures below 50 °C and within a pH range of 3–10 (Figure 1E,F).

### 3.2. Phage phiTH1 Recognizes CPS of K. pneumoniae as Its Receptor During Infection

CPS is one of the most common primary adsorption receptors for phages, facilitating their attachment to receptors located on the bacterial outer membrane [35]. To explore the role of CPS in mediating the susceptibility of *K. pneumoniae* to phage infection, we conducted a resistance screening experiment using the CPS-specific phage phiTH1 (Figure 2A). We successfully isolated five independent phage-resistant mutants of *K. pneumoniae* TH1: TH1_m1, TH1_m2, TH1_m3, TH1_m4, and TH1_m5. TEM analysis demonstrated that, compared to wild-type TH1, all five mutant strains completely lost the capsule layer (Figure 2B). Whole-genome sequencing further revealed that all resistant mutations were located within the *wcaJ* gene. The spectrum of mutations included insertions (TH1_m2), deletions (TH1_m4 and TH1_m5), and point mutations (TH1_m1 and TH1_m3), which were distributed across different regions of the gene sequence (Figure 2C). Subsequently, spot tests demonstrated that all five mutant strains acquired complete resistance to phiTH1 infection (Figure 2D). Consistent with the spot tests, phiTH1 was unable to lyse each mutant strain in liquid culture, and the cultures remained turbid (Figure 2D). We then complemented the *wcaJ* gene in each mutant strain. Spot tests showed that *wcaJ* complementation reinstated susceptibility to phiTH1, and the corresponding liquid cultures became clear, indicating successful phage-mediated lysis (Figure 2D).

Wza is responsible for transporting CPS from the periplasmic space to the bacterial cell surface. Therefore, we constructed a mutant strain lacking the *wza* gene to further verify the role of CPS in phage infection. TEM images showed that *wza* deletion resulted in the complete loss of CPS in the outer bacterial layer (Figure 2B). Spot tests revealed that phiTH1 formed lytic spots on *K. pneumoniae* TH1 but not on the Δ*wza* strain (Figure 2D). These results indicated that the K30-type CPS serves as an essential receptor for phiTH1 infection.

### 3.3. Genomic Characteristics of Phage phiTH1 and Prediction of Depolymerase

According to whole-genome sequencing, phiTH1 had a circular double-stranded DNA genome of 40,898 bp, with 52.97% G + C content and 52 coding DNA sequences (Figure 3A). Online BLASTn analysis of the phiTH1 genomic sequence revealed that phiTH1 was most closely related to phiW14/TH1-302-1 (MT431699.1), with 94% query coverage and 97.47% nucleotide identity, and *Klebsiella* phage YMR2 (PP662633.1), with 98% query coverage and 97.45% nucleotide identity (Table 1). Based on the comparative genomic and phylogenetic analyses, phage phiTH1 belongs to the genus *Przondovirus*, family *Autographiviridae*, and order *Caudovirales*, showing the highest similarity (91.768%) to *Klebsiella* phage vB_KpnP_KpV289 within the family (Figure 3B).

In the phiTH1 genome, the protein encoded by *orf5* was predicted to be a tail fiber protein. Sequence alignment revealed that the ORF5 protein has a high degree of sequence homology with various tail (fiber) proteins of other *Klebsiella* phages (Table 2). Moreover, phylogenetic tree results showed that they have close evolutionary relationship (Figure 3C). Among these proteins, the tail fiber protein ORF37 of *Klebsiella* phage K5-2 has been identified as a depolymerase specific for the K30/K69 capsule, with a similarity of 98.11% to ORF5, suggesting that ORF5 may have the same enzymatic activity. The tertiary structure prediction of the ORF5 protein revealed a pectate lyase domain (Figure 3D) which is generally associated with the recognition and degradation of polysaccharide substrates [36]. Furthermore, a comparison of the tertiary structures of ORF5 and ORF37 revealed that the two proteins are highly conserved in the core domain and secondary structure, but have obvious conformational variations in the N-terminal region, suggesting that the two proteins may have functional differences (Appendix A).

### 3.4. Identification of the Depolymerase Dop5, Encoded by the Orf5 Gene of Phage phiTH1

To verify CPS degradation activity of ORF5, recombinant His-tag protein encoded by *orf5* was successfully expressed and purified (Figure 4A). As expected, the recombinant depolymerase Dop5 formed translucent spots resembling plaque halos in K30-type (TH1, W14, and A552) *K. pneumoniae* but did not form spots on other K-type *K. pneumoniae*, such as K1-type (A2), K2-type (A25), K5-type (A23), K21-type (A66), K24-type (A512), K54-type (A1), and K64-type (A45) (Figure 4B). TEM images showed that the bacterial CPS structure disappeared after Dop5 treatment (Figure 4C). The measurement of uronic acid content also indicated a significant decrease in bacterial CPS following Dop5 treatment (Figure 4D). These results clearly demonstrated that Dop5 can degrade CPS. To determine whether capsule removal by Dop5 affects bacterial growth, we performed growth curve and time-dependent killing assays comparing *K. pneumoniae* TH1 with and without Dop5 treatment. Although Dop5 effectively degraded the bacterial capsule, it did not exhibit direct bactericidal activity (Figure 4E). We next evaluated whether Dop5 could affect bacteria that had developed phage phiTH1 resistance. When we applied purified Dop5 to lawns of the non-capsulated mutants and the wild-type TH1, a clear hydrolysis halo appeared only on the wild-type strain (Figure 4F). No activity was seen against the mutants, confirming that Dop5 depends on its specific K30 capsule substrate and cannot help overcoming phage resistance.

### 3.5. Protective Efficacy of Depolymerases Against K. pneumoniae Pulmonary Infection

This study established a murine pneumonia model using aerosolized inhalation of *K. pneumoniae* TH1 to evaluate the in vivo efficacy of the disaggregase Dop5 (Figure 5A). 100% mortality was observed in mice infected with TH1 aerosols within 7 days. Furthermore, high bacterial loads were detected in the blood and major organs of the mice, with the highest colony-forming unit (CFU) counts in the lungs (Figure 5B,C). The Dop5-treated group achieved an 80% survival rate, and bacterial loads in the blood and various organs were significantly lower than those in the untreated group (Figure 5B,C). C3 levels were significantly elevated in the untreated group compared to the control. However, Dop5 treatment significantly attenuated this infection-induced increase which can promote a normalization of C3 levels toward baseline (Figure 5D). Serum levels of inflammatory cytokines (IL-1β, IL-6 and TNF-α) in Dop5-treated mice were significantly lower than those in the untreated group and comparable to those in the control group (Figure 5E). These results suggest that Dop5 treatment can alleviate inflammatory responses and modulate complement system activation. Pathological examination of murine tissues revealed that untreated mice exhibited varying degrees of organ and tissue damage, including hydropic degeneration of hepatocytes, splenic congestion, hydropic renal tubular degeneration and renal interstitial congestion (Appendix A). And lung tissue showed marked thickening of alveolar walls, extensive neutrophil infiltration within the lung parenchyma, hemorrhagic signs and significantly increased lung pathology scores (Figure 5F,G). No significant histopathological damage was observed in the Dop5-treated group (Figure 5F,G). And The C3 level in the Dop5 group was higher than that in the control group, but this modest increase in C3 was not accompanied by any signs of tissue damage or inflammatory cytokine release. These results suggest that Dop5 exerts its protective effect by effectively reducing bacterial burden, thereby alleviating dysregulated immune activation and tissue damage.

## 4. Discussion

CPS is a critical virulence factor of pathogenic bacteria with a highly diverse structure [37]. To date, more than 80 serotypes have been identified in *K. pneumoniae* [16]. CPS can effectively resist phagocytosis and complement-mediated bactericidal mechanisms, and enhance the immune escape ability of bacteria [38,39]. By masking bacterial surface antigens, CPS reduces the deposition of complement components such as complement C3 and weakens the activation of classical pathway and alternative pathways to prevent the formation of the membrane attack complex (MAC), protecting bacteria from lysis and phagocytosis [40]. Furthermore, CPS can form a barrier structure to slow the penetration of antibiotics, which can reduce their therapeutic effects of antibiotics and enhance bacterial colonization and drug resistance [41]. CPS can also interfere with the interactions between host immune receptors and bacterial surface proteins [42]. A study has found that CPS can block *K. pneumoniae* recognition by the scavenger receptor LOX-1 to reduce macrophage-mediated uptake [43]; this mechanism enables the encapsulated strains to evade uptake and clearance by host immune cells, prolonging bacterial survival in the host. Accordingly, CPS is one of the main targets of various new targeted therapeutic strategies such as vaccines, monoclonal antibodies, phages, and their derived depolymerases. In this study, we isolated and identified a K30-type *K. pneumoniae* phage, phiTH1, and expressed the depolymerase, Dop5, which specifically degrades K30-type CPS.

However, frequent mutations in CPS loci are the main reason for the failure of phage treatment in *K. pneumoniae* [44]. CPS synthesis depends on the synergistic action of multiple glycosyltransferase genes, of which *wcaJ* and *wbaP* are key initiating glycosyltransferase genes [45]. The glycosyltransferase encoded by *wcaJ* transfers glucose-1-phosphate to the lipid carrier and initiates the synthesis of capsule repeat units. Loss of function of the *wcaJ* gene can prevent bacteria from synthesizing a complete capsule structure, presenting a non-capsulated phenotype. *K. pneumoniae* that has lost its CPS can develop phage resistance because phages cannot recognize and adsorb onto its surface [46]. The phage-resistant strains obtained in this study also showed capsule loss caused by *wcaJ* mutation, which is consistent with the above mechanism. Notably, a variety of mutational events were found in the *wcaJ* gene region, including frameshift insertions, deletions, nonsense mutations, and multiple missense mutations. Although the mutation types and locations varied, all variants resulted in consistent phenotypic consequences. These included loss of *wcaJ* glycosyltransferase function, complete bacterial capsule loss, and high resistance to phage phiTH1.

Structural alignment revealed that the catalytic domains maintain high consistency in spatial folding and distribution of key residues, but exhibit significant conformational differences in the N-terminal region, a localized variation that may significantly impact their function. Previous studies have shown that the N-terminus of phage tail fiber proteins is often responsible for recognition, assembly or anchoring to the tail base complex, while the C-terminal catalytic domain mediates specific recognition and degradation of polysaccharide substrates [11,47]. Therefore, slight conformational differences in Dop5’s N-terminal domain may lead to variations in host recognition or binding strength, potentially affecting the host range and lytic efficiency of the phage. Similar phenomena have been observed in other phage disaggregases. The tail protein of *K. pneumoniae* phage KP36 shares over 90% sequence similarity with closely related phages, but amino acid substitutions in just a few surface loops significantly alter its ability to infect strains with different K antigenic types [48].

Capsule depolymerase has dual functions: it helps bacteriophages recognize hosts to complete the process of adsorption and infection and can be used independently as an antibacterial tool [49]. Depolymerases have shown potential as antibacterial drugs against *K. pneumoniae*, such as Depo16 from phage vB_KpnP_ZK1, dep1011 from P1011, and Dep_ZX1 from phage vB_KpnP_ZX1 [50,51,52]. A study on mice with sepsis infected with hypervirulent K2-type *K. pneumoniae* demonstrated a synergistic effect between depolymerases and host immunity [30]. Our study showed that Dop5 did not have direct bactericidal activity in vitro; however, administration of the enzyme in a murine model of aerosol pneumonia significantly improved survival and reduced the bacterial load without causing histopathological damage. Although depolymerase does not directly kill bacteria, it can promote complement-mediated bactericidal effects and macrophage phagocytosis by removing the capsule and exposing bacterial surface antigens, demonstrating synergistic effects with the immune system [53]. The antimicrobial effects of phage therapy are primarily achieved through specific infection and lysis of target bacteria [5]. However, the emergence of phage-resistant mutants, endotoxin-induced inflammatory responses, unstable pharmacokinetics and a narrow host range of host immune neutralization have always been challenges faced during phage therapy [54,55]. Compared to phage therapies, the depolymerase-based therapeutic approach described in this study offers greater stability, predictable pharmacokinetics, lower risk of resistance development, easier production and purification. Given the potential of phage-derived Dop5 in antibacterial treatment, future research will further explore the synergistic mechanism between the Dop5 and the host immune system.

Protein-based therapeutics have been widely used to treat a variety of diseases, such as infections [56], tumors [57] and immune disorders [58]. However, studies have shown that protein-based therapeutics may trigger the production of anti-drug antibodies (ADAs), the release of proinflammatory cytokines or anaphylactic-like reactions, which can compromise efficacy, alter pharmacokinetics and induce immune-mediated toxicities [59,60]. The Dop5 described in this study shows significant therapeutic potential in mice, but as a protein-based biologic, its potential immunogenicity and safety require further evaluation. Future studies should focus on monitoring ADA production and its impact on efficacy after repeated dosing in animal models, measuring complement activation products (such as C3a and C5a) and inflammatory cytokine levels (such as IL-6, TNF-α, and IFN-γ) to assess the risk of immune overactivation, and conducting histopathological and toxicological analyses to identify potential side effects. These systematic assessments will lay the foundation for the safety validation and clinical translation of Dop5.

In summary, phage phiTH1 specifically infects K30-type *K. pneumoniae*, forming plaques with enlarged halos. Further research revealed that the depolymerase Dop5 derived from phiTH1 can specifically degrade the K30-type CPS. Although Dop5 does not directly kill the bacteria, it exhibits significant therapeutic potential in vivo. In a murine aspiration pneumonia model, Dop5 treatment effectively reduced infection severity. These findings indicate that the depolymerase is a type of highly promising antibacterial agent, especially suitable for treating infections caused by encapsulated bacteria that are difficult to deal with using traditional antibiotics.

## Figures and Tables

**Figure 1 viruses-17-01446-f001:**
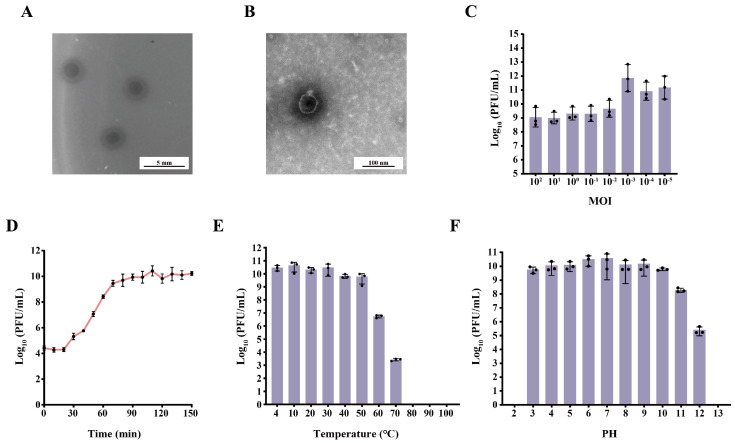
Morphology, biological characterization, and genomic features of phage phiTH1. (**A**) Plaques of phage phiTH1 formed on the double-layer agar plate. (**B**) Phage morphology observed with TEM. (**C**) Optimal MOI of phage phiTH1. (**D**) One-step growth curve of phage phiTH1. (**E**) Thermal stability of phage phiTH1. (**F**) PH stability of phage phiTH1. Data represent mean ± s.d. (*n* = 3).

**Figure 2 viruses-17-01446-f002:**
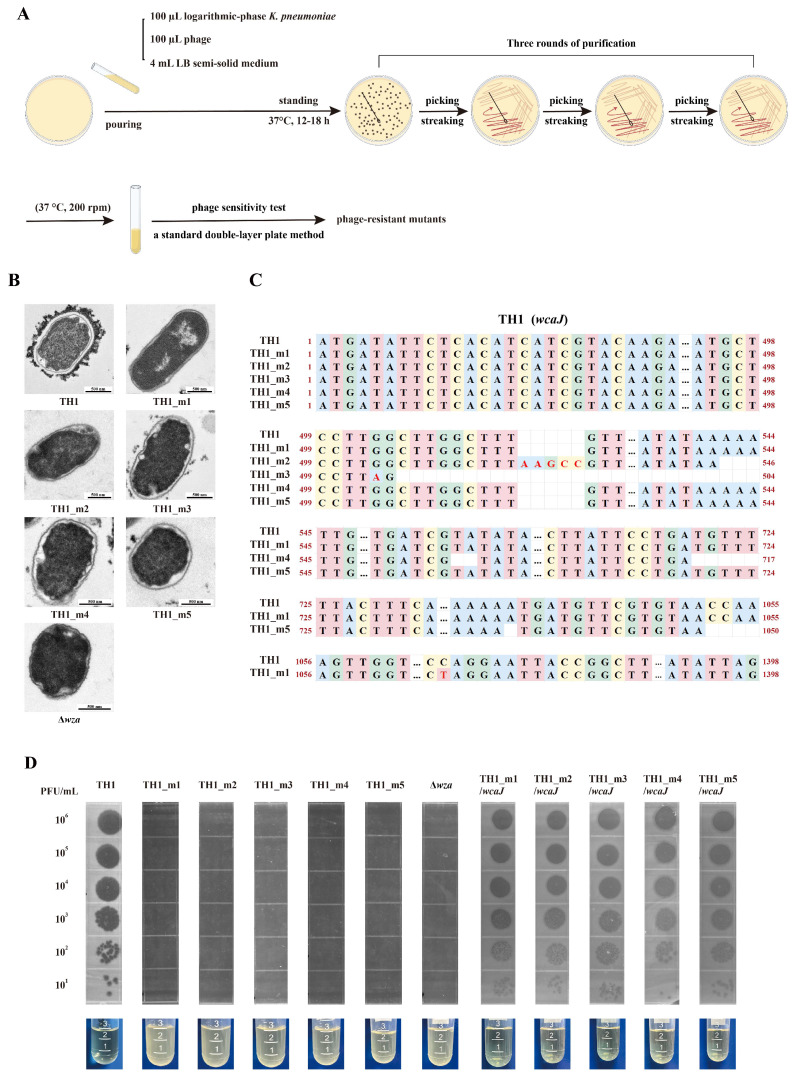
Identification and characterization of non-capsulated mutants and their phage susceptibility. (**A**) A schematic illustration of the rapid screening approach used to identify non-capsulated strains by figdraw.com. (**B**) TEM images of the wild-type TH1 strain, its five non-capsulated mutants (TH1_m1 to TH1_m5) and Δ*wza*. (**C**) Analysis of mutation sites in non-capsulated strains (TH1_m1 to TH1_m5). (**D**) Spot tests were performed by applying serial dilutions (from 10^1^ to 10^6^ PFU/mL) of phage phiTH1 onto bacterial lawns of different strains to assess phage susceptibility. Corresponding liquid culture assays were conducted to observe bacterial lysis.

**Figure 3 viruses-17-01446-f003:**
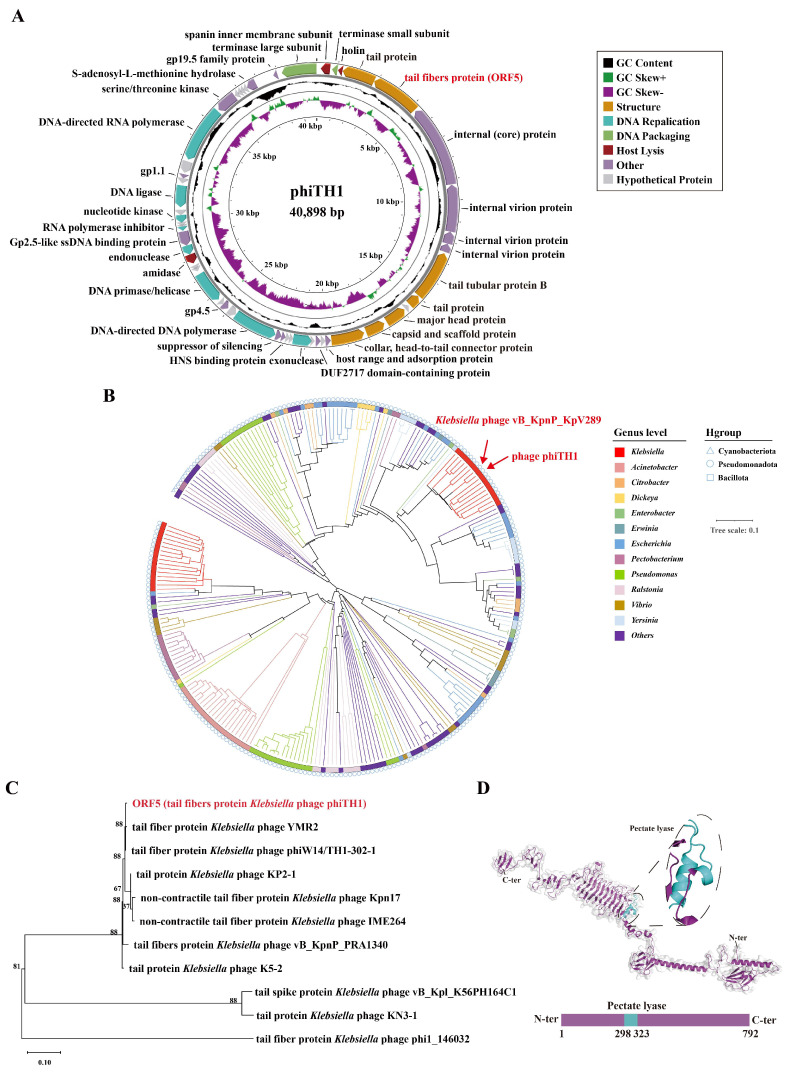
Identification and characterization of phage phiTH1 and its putative depolymerase. (**A**) The genomic map of phage phiTH1. (**B**) Phylogenetic tree of phiTH1 and related *Autographiviridae* phages. (**C**) The Maximum Likelihood method with bootstrap testing (1000 replicates) was applied for the construction of phylogenetic trees based on gene sequences of the predicted depolymerase. (**D**) Protein three-dimensional structure prediction of the predicted depolymerase and protein domain.

**Figure 4 viruses-17-01446-f004:**
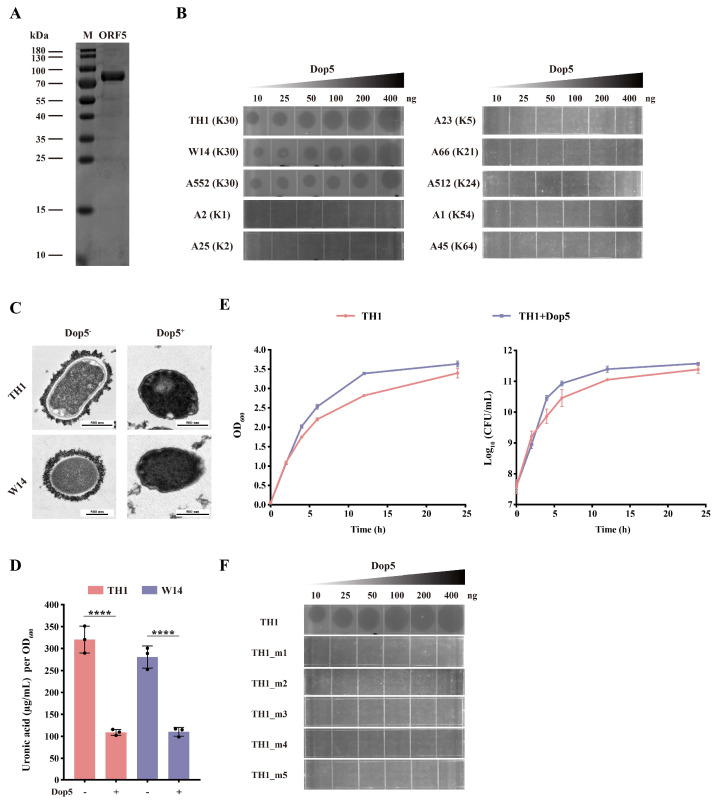
Biological activity of the depolymerase Dop5. (**A**) The purified ORF5 protein was separated on SDS-PAGE with coomassie blue staining, and the molecular weights were indicated beside the protein markers. (**B**) Depolymerase activity of Dop5 was detected when spotted on K30-type *K. pneumoniae* lawns, but not on other K-type *K. pneumoniae*. (**C**) TEM of K30-type *K. pneumoniae* (TH1 and W14) treated with or without depolymerase Dop5. (**D**) The measurement of uronic acid content of K30-type *K. pneumoniae* (TH1 and W14) treated with or without depolymerase Dop5. Data represent mean ± s.d. (*n* = 3). Statistical significance was determined using Student’s *t*-test (**** *p* < 0.0001). (**E**) The 24 h growth curves and 24 h time killing kinetics of *K. pneumoniae* TH1 treated with or without depolymerase Dop5. (**F**) Detection of depolymerase Dop5 activity against *K. pneumoniae* TH1 and non-capsulated mutants (TH1_m1 to TH1_m5).

**Figure 5 viruses-17-01446-f005:**
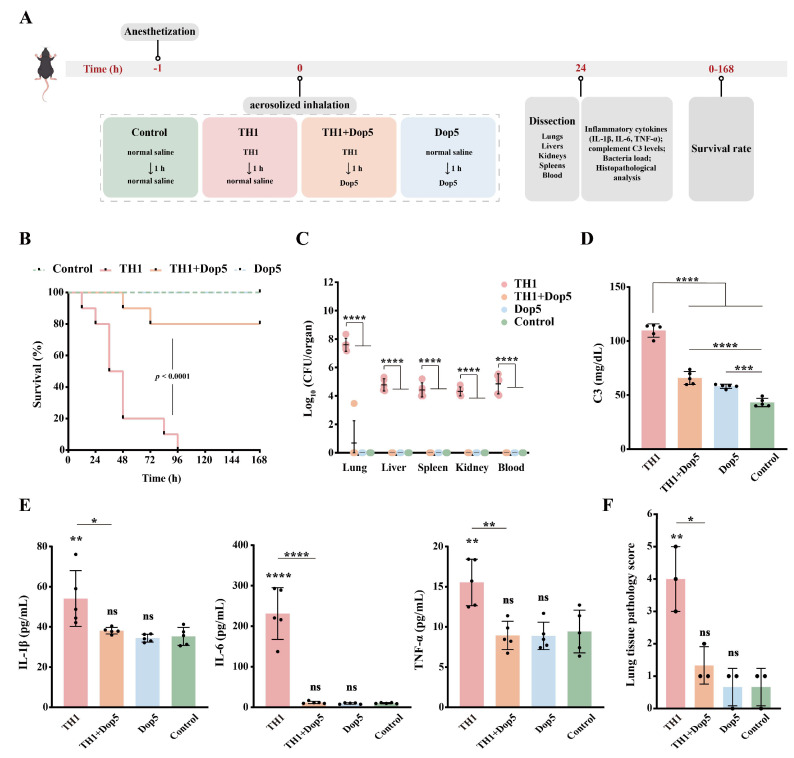
Therapeutic efficacy of Dop5 in a murine aspiration pneumonia model. (**A**) The experimental flowchart of the murine pneumonia model by figdraw.com. (**B**) Survival of mice (*n* = 10) following aerosol inhalation with TH1, TH1 + Dop5, Dop5, and the control group. Statistical significance was analyzed using the Log-rank (Mantel–Cox) test. (**C**) Bacterial load counts in the blood and organs of mice in different groups. Data represent mean ± s.d. (*n* = 5). Statistical significance was analyzed by two-way ANOVA (**** *p* < 0.0001). (**D**,**E**) Levels of proinflammatory cytokines IL-1β, IL-6, and TNF-α, and complement C3 in serum at 24 h post-infection. Data represent mean ± s.d. (*n* = 5). Statistical significance was analyzed by one-way ANOVA (* *p* < 0.05; ** *p* < 0.01; *** *p* < 0.001; **** *p* < 0.0001; ns, not significant). (**F**) Histopathological scores of lung tissue sections. Data represent mean ± s.d. (*n* = 3). Statistical significance was analyzed by one-way ANOVA (* *p* < 0.05; ** *p* < 0.01; ns, not significant). (**G**) Anatomy and H&E staining (200× and 400× magnifications) of lungs from mice in different groups. Neutrophil infiltrations are indicated by red arrows.

**Table 1 viruses-17-01446-t001:** High genome identity and query coverage between *Klebsiella* phage phiTH1 and closely related phages.

Scientific Name	Query Coverage	Identity	Length
*Klebsiella* phage phiTH1	100%	100%	40,898
*Klebsiella* phage YMR2	98%	97.45%	40,756
*Klebsiella* phage phiW14/TH1-302-1	94%	97.47%	39,762
*Klebsiella* phage IME264	93%	94.38%	40,671
*Klebsiella* phage phiA25	95%	94.6%	41,030
*Klebsiella* phage P293	95%	96.21%	41,968
*Klebsiella* phage RCIP0210	93%	97.05%	39,631
*Klebsiella* phage IME183	92%	94.41%	41,384
*Klebsiella* phage vB_Kp_IME531	90%	96.18%	40,314
*Klebsiella* phage K5-2	91%	92.94%	41,116

**Table 2 viruses-17-01446-t002:** High sequence identity of depolymerase Dop5 to homologous tail (fiber) proteins.

Scientific Name	Query Coverage	Identity	Length
tail fibers protein [*Klebsiella* phage phiTH1]	100%	100%	792
tail fiber protein [*Klebsiella* phage phiW14/TH1-302-1]	100%	99.62%	792
tail fiber protein [*Klebsiella* phage YMR2]	100%	99.37%	793
tail fibers protein [*Klebsiella* phage vB_Kp_IME531]	100%	98.11%	792
tail protein [*Klebsiella* phage K5-2]	100%	98.11%	792
tail protein [*Klebsiella* phage VLCpiA3b]	100%	97.85%	792
non-contractile tail fiber protein [*Klebsiella* phage phiA25]	100%	97.6%	792
tail fibers protein [*Klebsiella* phage vB_KpnP_PRA1340]	100%	97.35%	792
non-contractile tail fiber protein [*Klebsiella* phage IME264]	100%	96.59%	792
tail protein [*Klebsiella* phage Whistle]	100%	96.72%	792

## Data Availability

The whole-genome sequencing data for strains W14 and TH1 have been deposited in the NCBI database under accession numbers [NZ_CP015753.1] and [NZ_CP016159.1]. The annotated whole-genome sequences of phages phiTH1 can be found at NCBI with accession number PP801331. The raw whole-genome sequencing data for the wild-type TH1 strain and the derived mutant strains (TH1_m1 to TH1_m5) have been deposited in the SRA under BioProject accession number PRJNA1315033.

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
