# Peer review of "Characterization of a Phage-Encoded Depolymerase Against *Klebsiella pneumoniae* K30 Capsular Type and Its Therapeutic Application in a Murine Model of Aspiration Pneumonia"

_viruses, 2025, doi:10.3390/v17111446_

Round 1

Reviewer 1 Report (Previous Reviewer 3)

Comments and Suggestions for Authors

The authors have done a good job and significantly improved the manuscript.

Author Response

We thank the reviewer for the valuable feedback and the final recommendation for publication.

Reviewer 2 Report (Previous Reviewer 2)

Comments and Suggestions for Authors

The authors of the manuscript “Identification of a Novel Phage-Encoded Depolymerase against Klebsiella Pneumoniae K30 Capsular Type and Its Therapeutic Application in a Murine Model of Aspiration Pneumonia” isolated and studied a lytic bacteriophage called phiTH1, which targets Klebsiella pneumoniae, and sequenced its genome. They discovered that the phage infects its host by binding to the K30-type capsular polysaccharide. A tail fiber protein named Dop5, containing a pectate lyase domain, was identified as a likely depolymerase for the K30-type capsule. The team then produced and purified the recombinant Dop5 protein in Escherichia coli and confirmed its ability to break down the capsule in laboratory tests. Using a mouse model of aspiration pneumonia caused by K30-type Klebsiella pneumoniae TH1, they demonstrated that Dop5 provided significant protection, saving 80% of the mice from a fatal infection. These findings highlight the therapeutic potential of Dop5 as a new treatment for Klebsiella pneumoniae infections.

The manuscript is well written and summarizes a large amount of scientific information. The authors took into account all the comments of the reviewers and did a lot of work on editing the manuscript in accordance with the comments of the reviewers. This article can be published in Viruses

Author Response

We thank the reviewer for the valuable feedback and the final recommendation for publication.

Reviewer 3 Report (Previous Reviewer 1)

Comments and Suggestions for Authors

In this study, the authors identify and characterize a novel phage-derived depolymerase, Dop5, and demonstrate its therapeutic efficacy against K30-type Klebsiella pneumoniae in a murine pneumonia model.

The authors have responded to the previous review comments and have made corresponding revisions and supplements to the manuscript. These revisions have improved the completeness and clarity of the study to a certain extent. However, the following minor issues still require further attention and polishing before the manuscript can be considered for final acceptance:

  1. The tail fibers in Figure 1B require clearer visualization. Provide a morphological comparison with related phages and include the full CsCl purification protocol (gradient, speed, duration) in Methods.
  2. To validate the definitive statement regarding the absence of non-wcaJ mutations, the methodological details must be provided, including the specific bioinformatics tools, sequencing coverage, and variant filtering criteria applied in the genomic analysis.
  3. In Figure 3B, the phylogenetic scope of the VIPtree analysis appears overly broad, obscuring the precise position of phiTH1 within the Autographiviridae family and its relationship to closely related phages. It is recommended to regenerate this figure focusing on phiTH1's specific viral family and relevant host groups.
  4. As C3 is a positive acute-phase protein, its increase during infection is plausible. The effect of Dop5 treatment might be more accurately described as promoting a normalization or restoration toward baseline levels rather than a simple "reduction," which could be misinterpreted in the context of complement consumption.
  5. Figure 5C indicates that bacterial loads in organs were nearly zero after Dop5 treatment. As complete bacterial eradication is unlikely, please provide details in the Materials and Methods regarding the colony counting methodology, specifically the lower limit of detection.
  6. In the Materials and Methods section describing the animal model, please briefly state the rationale for the selected Dop5 dose (e.g., 50 µg). Was this based on preliminary efficacy dataor pharmacokinetic studies?
  7. For Figure 5F, please add labels or a legend to the x-axis to clearly indicate the experimental groups being compared.
  8. Please specify the experimental method used to quantify the serum levels of IL-1β, IL-6, TNF-α, and C3 in the Materials and Methods.
  9. In Figure 5E, the y-axis label "TNF-a" should be corrected to the standard symbol "TNF-α".
  10. A formatting issue is noted on Line 58, where a PMID appears to be incorrectly cited. Please carefully check all citations throughout the manuscript.

Author Response

This manuscript is a resubmission of an earlier submission. The following is a list of the peer review reports and author responses from that submission.

Round 1

Reviewer 1 Report

Comments and Suggestions for Authors

This study identified a phage-derived depolymerase, Dop5, which demonstrated protective efficacy against K30-type Klebsiella pneumoniae infection in a murine aspiration pneumonia model. The work includes detailed phage characterization and comprehensive evaluation of depolymerase antibacterial activity, presenting a well-rounded study with promising clinical potential. However, improvements in certain experimental methods and result analyses would strengthen the manuscript and make it more suitable for publication.

Major comments to the authors:

  1. The TEM images in Figure 1 are of suboptimal quality, making it difficult to clearly visualize key tail fiber structures that may be relevant for the depolymerase activity. The authors are advised to purify the phage particles using cesium chloride density-gradient ultracentrifugation prior to imaging to improve resolution.
  2. In Figure 2A, the workflow for isolating phage-resistant mutants is insufficiently described. Please clarify whether multiple rounds of purification were performed to ensure stable resistance, and include the detailed procedure in the Materials and Methods section.
  3. The sequencing information for TH1_m1 to TH1_m5 is incomplete. Additional discussion on the randomness and diversity of these mutants is necessary. Beyond wcaJ, were other mutations identified? Are there notable phenotypic or genetic differences among the five resistant strains?
  4. In Figure 4B, the plaque morphology of strains TH1 and W14 differs considerably, with W14 showing less clear halos. However, in Figure 4D, both strains appear to have similar CPS degradation levels. This discrepancy requires further clarification.
  5. The manuscript lacks essential information about strain W14. What is the genetic background of W14, and why was it included in the depolymerase activity assays? How does it differ from strain TH1?
  6. The conclusion that Dop5 specifically targets K30-type pneumoniaerequires stronger evidence. The authors are encouraged to include additional K30 clinical or reference strains, as well as representative strains of other capsular types, to comprehensively evaluate the specificity and efficiency of Dop5.
  7. It remains unclear whether Dop5 exerts any effect on capsule-deficient mutants (TH1_m1to TH1_m5). Testing this could provide insights into whether depolymerases may help overcome phage resistance mechanisms.
  8. In the mouse pneumonia model, the therapeutic benefit of Dop5 is highlighted, but no direct comparison with phage treatment alone is provided. Considering that phages have bactericidal effects while depolymerases act indirectly, it would strengthen the study to discuss whether Dop5 indeed confers superior therapeutic outcomes compared with phage administration.
  9. The study shows that Dop5 protected 80% of mice; however, mechanistic explanations are lacking. Is this due to enhanced bacterial clearance, improved immune recognition, or other effects? Additional measurements (e.g., neutrophil infiltration, cytokine levels, complement deposition) would help elucidate Dop5’s therapeutic mechanism and unique advantages.
  10. Protein-based therapeutics often face challenges such as immunogenicity or potential side effects, but these aspects are not discussed. It is recommended to address the possible immunogenicity of Dop5, its implications for clinical application, and outline future directions for safety evaluation.

Reviewer 2 Report

Comments and Suggestions for Authors

The authors of the manuscript “Identification of a Novel Phage-Encoded Depolymerase against Klebsiella Pneumoniae K30 Capsular Type and Its Therapeutic Application in a Murine Model of Aspiration Pneumonia” isolated and studied a lytic bacteriophage called phiTH1, which targets Klebsiella pneumoniae, and sequenced its genome. They discovered that the phage infects its host by binding to the K30-type capsular polysaccharide. A tail fiber protein named Dop5, containing a pectate lyase domain, was identified as a likely depolymerase for the K30-type capsule. The team then produced and purified the recombinant Dop5 protein in Escherichia coli and confirmed its ability to break down the capsule in laboratory tests. Using a mouse model of aspiration pneumonia caused by K30-type Klebsiella pneumoniae TH1, they demonstrated that Dop5 provided significant protection, saving 80% of the mice from a fatal infection. These findings highlight the therapeutic potential of Dop5 as a new treatment for Klebsiella pneumoniae infections.

The authors have done a good scientific study. However, the question arises about the novelty of the results obtained. If the authors had compared their results with the data described in the literature, they would have found that an almost identical depolymerase of the K5-2 phage was studied in 2017 (Hsieh, P. F., Lin, H. H., Lin, T. L., Chen, Y. Y., & Wang, J. T. (2017). Two T7-like Bacteriophages, K5-2 and K5-4, Each Encodes Two Capsule Depolymerases: Isolation and Functional Characterization. Scientific reports, 7(1), 4624. https://doi.org/10.1038/s41598-017-04644-2). Comparison of proteins of two phages leads to the following results: Percent identity: 98.11%, Percent similarity: 98.74%.

Hsieh et al isolated and characterized two Klebsiella bacteriophages, K5-2 and K5-4, capable of infecting strains with capsular types K30/K69 and K5 or K8 and K5, respectively. Each phage carried two open reading frames (ORFs) encoding putative capsule depolymerases. The first ORF produced tail fiber proteins with depolymerase activity against K30/K69 or K8 capsules, while the second ORF encoded nearly identical hypothetical proteins with K5 depolymerase activity. In vitro experiments using Alcian blue staining confirmed that the purified depolymerases cleaved Klebsiella capsular polysaccharides (CPS), releasing monosaccharides. Phage infection depended on the presence of the K5 capsule, as deletion mutants resisted lysis. While the phages effectively killed bacterial strains, purified depolymerases alone did not exhibit bactericidal effects. These findings demonstrate that both phages and their depolymerases specifically target K30/K69, K8, or K5 capsules, highlighting their potential for bacterial typing and therapeutic applications against K. pneumoniae infections. Hsieh et al tested K5-2 and K5-4  phages/depolymerases on a large number of K-types. The authors (Chen et al) only checked that the phiTH1phage does not work on the K1 and K2 types.

The authors did not check whether their depolymerase differs from well-studied proteins. I guess not. I do not find any scientific novelty in the presented research and do not recommend it for publication.

Reviewer 3 Report

Comments and Suggestions for Authors

The manuscript written by Y. Chen and co-authors is a completed study describing the isolation and characterization of a Klebsiella bacteriophage, as well as the preparation and characterization of the depolymerase of this phage. The phage is specific to K30-type Klebsiella pneumonia and its depolymerase Dop5 can protect mice infected with the K. pneumonia strain, which is a host strain for the phage. Overall, this study proves the potential of depolymerase as a possible antimicrobial agent.

However, there are a number of doubts about the novelty of the presented research.

The authors indicate in the title “Identification of a Novel Phage-Encoded Depolymerase”. However, the article published by Hsieh PF et al “Two T7-like Bacteriophages, K5-2 and K5-4, Each Encodes Two Capsule Depolymerases: Isolation and Functional Characterization. Scientific reports, 2017, 7:4624 (https://doi.org/10.1038/s41598-017-04644-2) describes phage depolymerases, one of which has a remarkably high amino acid sequence similarity to Dop5 and can lyse K30 type K. pneumonia capsules. There is no reference to this study in the peer-reviewed manuscript and, therefore, there is no comparative analysis of aa sequences and their enzymatic properties (the similar depolymerase destroys the K30 and K5 capsules).

If the authors conduct a more thorough comparative analysis of the phiTH1 genome (Phylogenetic tree acquired using the ViPTree program, Comparative genome alignment performed in ViPTree, VIRIDIC, etc.), they would have found similar phages. However, comparative genome analysis is not shown.

An important part of this study is in vivo experiments. It is this part of the work that has a novelty. However, the authors described this part of the work too briefly. Section 2.15 does not describe the choice of the infectious dose, only mentions "preliminary experiments". 60 animals were used, but only 3 groups of 10 mice were described.

I believe that the manuscript needs Major revision:

  • More detailed comparative phage genome analysis.
  • Comparison of sequencesand properties of the Dop 5 with similar, previously described depolymerases. This would improve the Discussion section.
  • A more thorough description of the in vivo experiments and their results.
